# Observation of Traditional Caregiver-Infant Feeding Behaviours and Porridge and Energy Intakes during One Meal to Define Key Messages for Promoting Responsive Feeding in the Amparafaravola District, Rural Madagascar

**DOI:** 10.3390/nu14020361

**Published:** 2022-01-15

**Authors:** Yannick Razafindratsima, Andrimampionona Razakandrainy, Sonia Fortin, Charlotte Ralison, Claire Mouquet-Rivier

**Affiliations:** 1Department of Food and Nutrition Science, University of Antananarivo, P.O. Box 175, Antananarivo 101, Madagascar; razafindratsima.mg@gret.org (Y.R.); ralisoncharlotte@yahoo.com (C.R.); 2GRET (Professionnels du Développement Solidaire), BP 1563, Antananarivo 101, Madagascar; razakandrainy.mg@gret.org; 3Qualisud, University of Montpellier, Avignon University, CIRAD, Institut Agro, IRD, Université de la Réunion, 34394 Montpellier, France; sonia.fortin@ird.fr

**Keywords:** complementary food, appetite, counselling card, undernutrition

## Abstract

Undernutrition is highly prevalent in young children in Madagascar and insufficient intake per meal could be one of the main causes. A cross-sectional survey of infant feeding practices including video-recorded meal observations was carried out with 101 caregiver–infant pairs in the Amparafaravola district, Northeast Madagascar. The objective was to quantify the porridge/energy intake of 9–11-month-old children and assess its association with the caregiver–infant feeding behaviours. Then, key messages for promoting responsive feeding (RF) were developed and tested through focus group discussions. The mean porridge intake was 12.8 ± 7.5 g/kg body weight (BW)/meal, corresponding to hardly one-third of the 300 kcal recommended from complementary foods for 9–11-month-old children. Analysis of meal videos suggested that mothers practiced the five positive feeding behaviours (self-feeding, responsive, active, social, and distraction), and rarely the negative ones. Only 6.9% of mothers used positive RF “very frequently”, although it was associated with higher intakes (*p* < 0.05), with mean intake reaching 21 g/kg BW. In focus groups, caregivers approved the six RF messages and related counselling cards. They suggested some modifications to improve their understanding, and counselling cards were revised accordingly. The long-term impact of RF-promoting card use on the meal intakes and the nutritional status of young children must now be assessed.

## 1. Introduction

The first thousand days of life are a critical period for children’s development and growth, commonly seen as “a window of opportunity” for implementing intervention strategies. Inadequate feeding during this period, among other factors, may lead to malnutrition with irreversible consequences on children’s health status and development [1,2]. In 2020, the global stunting prevalence among <5-year-old children was 22% [3], and the situation in sub-Saharan Africa was even more alarming. In Madagascar, reducing undernutrition is a big challenge due to the very high prevalence of stunting (42%), one of the highest in the world [4], and also anaemia (approximately 50%) [5] in this age group. In the Alaotra Mangoro region, known as the rice barn of Madagascar, 47% of <5-year-old children present stunted growth [4] and close to 50% have anaemia [5].

Childhood undernutrition has multiple causes, including inadequate food intake, insufficient in energy, macronutrients and micronutrients [1,2,6]. Several factors, related to household, caregivers, children and the nutritional or sensory quality of complementary foods, may lead to insufficient nutritional intake from complementary foods in young children [7]. According to studies carried out in several countries in sub-Saharan Africa [8,9,10], the amount of complementary food ingested by young children during a meal is low, well below the gastric capacity of 30 g/kg body weight (BW)/meal [7]. Appetite plays also an important role in food intake. The problem of “small appetite” or anorexia in young children is frequently reported by mothers in unfavourable contexts [11]. Since 1980, the frequent presence of some symptoms linked to respiratory diseases or diarrhoea has been associated with appetite loss in preschool children [12]. Moreover, repeated exposure to pathogenic microorganisms, due to poor environmental and domestic hygiene, can lead to a chronic inflammatory condition of the intestine, initially called tropical enteropathy and then environmental enteric dysfunction (EED) [13,14]. EED reduces nutrient absorption, due to the intestinal mucosa inflammation, and is also associated with decreased appetite [14]. In addition, the use of unbalanced complementary foods, characterised by low energy and nutrient densities, worsens the low nutritional intake by young children.

Different strategies can be used to increase the energy and nutrient intake by young children. For instance, the use of high energy- and nutrient-dense porridges, obtained thanks to partial starch hydrolysis [15,16] and fortification [17] during the manufacture of complementary foods. Regarding the organoleptic properties, a study in Burkina Faso showed a positive effect on porridge intake by young children when the porridge sweetness was improved by increasing sucrose content from 1 to 9 g/100 g dry matter (DM) [18]. Several of these strategies have been adopted in Madagascar to develop Koba Aina™ (Antananarivo, Madagascar), a locally processed fortified cereal-based blend for use as complementary food for 6-23-month-old children. This food product, marketed by the social enterprise Nutrizaza (Antananarivo, Madagascar) since 2011, is the fruit of research carried out by (GRET), French National Research Institute for Sustainable Development(IRD) and the Laboratory of Biochemistry Applied to the Food Sciences and Nutrition(LABASAN, University of Antananarivo). The food fortification project PFOA funded by the European Union (2017–2021) was then implemented in Madagascar to improve the accessibility of this fortified complementary food with the aim of preventing various forms of malnutrition, such as anaemia and micronutrient deficiencies. However, although the porridge prepared from this product has nutrient densities in accordance with the main international recommendations [19,20,21], it will allow meeting the nutritional needs only if consumed in sufficient quantity every day. Therefore, it was important also to quantify the amount of porridge eaten by young children and to find strategies to increase their porridge intake. Among these strategies is the implementation of “responsive feeding” (RF), a feeding style to promote positive interaction between the caregiver (usually the mother) and the child, which will help the child learn to self-regulate his or her feeding and, in settings where undernutrition is highly prevalent, stimulate food intake [22,23,24]. The practice of RF, which gives importance to how a child is fed, has been recommended by the WHO [25,26] for a long time, but most studies have been performed in high-income countries where overweight is the major concern [24]. So far, existing studies to promote this feeding style [27,28] have not been interested in the precise measurement of the quantity of complementary food ingested by children. Recently, there have been increasing efforts to better assess feeding practices, including the use of RF, with proposed indicators and their relationship to intake assessed in different ways, either by direct observation, by weighing or counting bites, or by quantitative 24-h recall [10,29,30]. A study in Burkina Faso showed that encouraging the child to eat has a significant positive effect on food intake [8]. A study in Ethiopia [10] found that some caregiver–child pairs’ feeding behaviours during a meal are associated with the energy intake of 9–11-month-old children. Another study on slightly older children (12–23 months) in a rural area in Ethiopia showed that the implementation of RF practices is associated with a higher number of bites accepted by the children [29]. Nevertheless, data are still scarce. Showing and quantifying how the responsive feeding is practiced by communities, and how much it can influence food and nutritional intakes remains important in contexts where child undernutrition is widespread and may constitute an additional argument in favour of this practice.

The present study was carried out in the north-eastern part of Madagascar, in the predominantly rural district of Amparafaravola in the Alaotra mangoro region. It consisted of two stages. First, an observational cross-sectional study was performed in 101 children (9–11 months of age) to measure their intake of an energy-dense fortified porridge meal and to characterise caregiver–child feeding styles in order to identify behaviours with positive effects on food intake. Second, contextually adapted messages and the corresponding promotion tools were designed to promote RF and increase the amount of porridge consumed by children. Finally, the understanding, feasibility and acceptability of these key messages and tool were tested in focus groups in the community.

## 2. Subjects, Materials and Methods

### 2.1. Ethical Aspects

The study protocol was approved by the ethical committee for biomedical research of Madagascar (reference: 100-MSANP/CERBM), under the Ministry of Public Health, on 20 October 2017.

### 2.2. Observational Study to Characterise Traditional Feeding Styles and Practices; Intake Measurement

#### 2.2.1. Study Design, Location and Participants

The observational cross-sectional study on 101 caregiver–child pairs was carried out in six villages of the Amparafaravola district in 2018. The inclusion criteria for the infant population were: 9–11 months old, breastfed, to have started to consume complementary foods at the survey time and not to be severely wasted (Z-score weight-for-length > −3). The inclusion of caregivers, primarily the mothers when present, was not based on specific inclusion criteria, other than agreeing to participate in the study. Feeding behaviours were observed during one meal and food intake was measured by weighing. Anthropometric measurements, socio-economic data, and data on feeding practices were collected. After trained field workers explained the study course and objective to the parents and left them an information sheet, the informed consent was signed by the child’s caregivers, mainly represented by the mother.

#### 2.2.2. Characteristics of the Porridge Used in the Study

All meal observations were made with the same food, i.e., a porridge prepared with the fortified cereal-based blend Koba Aina™ (Antananarivo, Madagascar), produced locally. This product is mainly composed of local ingredients, fortified with the addition of a mineral and vitamin premix, and formulated to meet the macro- and micro-nutrient requirements of young children when used as recommended on the package. The product also contains a small amount of BAN™ (Bagsværd, Danemark) amylase to enable its preparation at the concentration of 25 g of DM/100 g porridge, corresponding to an energy density of 100 kcal/100 g porridge, with a consistency suitable for young children. 

#### 2.2.3. Meal Observations and Intake Measurements

An appointment was made the day before the observation to fix the visit time at each household the next morning. Caregivers were kindly asked not to breastfeed/feed their child in the hour before the appointment. 

The child’s food intake during the study meal was measured by weighing the plate before and after consumption using kitchen scales (precision 1 g), by taking into account losses during consumption. Assessors prepared the porridge at the children’s home according to a standardised procedure. Two sachets of 35 g of cereal-based blend with the amount of water needed (210 g) were used to obtain approximately 260 g of porridge with a DM content of 25 g/100 g to allow ad libitum eating. Aliquots of approximately 15 g of the porridge were taken and stored in an icebox to measure the DM content of the prepared porridge meal. In the laboratory, the DM content was determined in triplicate by oven-drying at 104 °C until constant weight. Then, the porridge intake was calculated as DM that was converted to energy using the energy value of the cereal-based blend (400 kcal/100 g DM).

Each study meal was video-recorded to analyse the caregiver–child pairs’ feeding behaviours. Assessors were asked to position themselves as discreetly as possible and not to speak with the mother, the child or any other person during the meal. Mothers were asked not to change their usual way of feeding their child.

#### 2.2.4. Meal Video-Recording Analysis

The 101 videos were analysed twice by two different trained assessors (i.e., four times in total) using the method described by Abebe et al., 2017 [29] and Baye et al., 2018 [10] in Ethiopia, with modifications. Feeding behaviours were coded using the five categories of behaviours proposed by Moore (2006) [31]: self-feeding, responsive feeding (RF), active-feeding, social behaviour, and distraction. Assessors rated the behaviour as positive when it increased and as negative when it decreased porridge intake by children. Examples of each behaviour type are in the supplemental data by Baye et al. (2018) [10]. To characterise the feeding style, standardised scores for the frequency of the observed behaviours were established as follows: a behaviour was considered absent when never observed or only once (standardised score = 0), present when observed two to four times (standardised score = 1), and frequent when observed more than four times (standardised score = 2). An additional analysis was done when the scores attributed by the two assessors strongly differed. Then global positive and negative scores were built by summing the standardised scores of the five behaviour categories, for the mothers and for the children. Scores were averaged and the mean scores were used for the statistical analysis. Breastfeeding during the meal was counted. During the video analysis, the two assessors also empirically evaluated the overall feeding style according to the following types: RF, active positive, active negative (controlling), laissez faire [10].

#### 2.2.5. Anthropometric Measurements

Each child’s weight and length were measured using standardised methods recommended by WHO [32]: weight in light clothing was determined using the double-weighing method (precision 0.1 kg), and length was measured in supine position using a wooden measuring board (precision 1 mm).

#### 2.2.6. Socio-Economical Characteristics and Feeding Practices

A pre-tested questionnaire was used to collect information from caregivers on the household socioeconomic and demographic situation (e.g., goods owned, housing quality, caregivers’ age, education level and occupation, child’s date of birth, sex, birth order), as well as on breastfeeding and complementary feeding practices from the child’s birth until the survey day. To calculate the child’s dietary diversity score, caregivers were also asked whether their child had eaten foods from fourteen different food groups on the day before the survey. The fourteen food groups were then grouped into seven food groups following the methodology recommended by WHO [33,34]. Finally, after the study meal, caregivers were asked to describe their child’s usual appetite and on the observation day. 

#### 2.2.7. Data Management and Statistical Analysis

Data on the study meal, including quality checks and validation by double-entry, were entered using the Epidata 3.1 (Odense, Denmark) software. Statistical analyses were performed using Epidata Analysis 1.6 (Odense, Denmark) and Statgraphics Centurion XVI.II (Warrenton, Virginia). 

Some indices and scores were created to analyse the collected data. The household socioeconomic level was assessed with an index constructed from the responses related to property owned (12 variables) and housing quality (2 variables). Then, this index was categorised in terciles: low, medium, and high socioeconomic level.

Weight-for-length, length-for-age and weight-for-age Z-scores were calculated with the Anthro software [35]. The diet diversity score was calculated based on eight food groups, i.e., seven food groups plus breastmilk, as recommended by the WHO et al. (2008; 2010; 2021) [33,34,36].

Qualitative variables were expressed as frequencies, and continuous variables as means (standard deviation, SD). Crude associations of the amount of porridge consumed (g/kg BW/meal) and energy intake (kcal/day) with the socioeconomic, anthropometric and behavioural variables were assessed using one-way analysis of variance (ANOVA) for qualitative factors, and univariate linear regressions for quantitative variables. Next, generalised adjusted linear models were used to explore the associations with the different observed behaviours, including variables found to be significantly associated with the children’s food and energy intake in univariate analysis. In all analyses, associations were considered statistically significant for *p*-values < 0.05. 

### 2.3. Validation of Key Messages for Responsive Feeding Promotion in the Amparafaravola District

#### 2.3.1. Definition of the Key Messages and Creation of an Awareness-Raising Tool for Their Promotion

Based on the analysis of the mealtime videos, and using the recommendations of the Pan American Health Organisation (PAHO) [24] and Moore et al. [31], key RF-promoting messages were developed with the aim of improving the dietary intake of 9–11 month-old children in Amparafaravola district. Then, counselling cards were designed to illustrate each message with the support of the GRET communication team in Madagascar, which has experience in graphic design in the field of public health nutrition. For the formulation of each message and the choice of the corresponding practices, only affirmative sentences (without “not to do” injunctions and prohibition crosses) were used, in accordance with the Madagascar National Office of Nutrition recommendations. 

#### 2.3.2. Assessment of the Key Message through Focus Group Discussion

A qualitative study based on focus group discussions (FGD) was performed in three municipalities of the Amparafaravola district to assess the perception of the RF messages and the corresponding counselling cards, developed as awareness-raising tools. The FGDs were performed with four participant types: mothers, fathers, other caregivers (grandmothers, aunts…) and community health workers (CHW). The main recruitment criterion was to reside or work at the survey sites. Age, sex, socio-professional category and education level were recorded. Participants were informed about the FGD course and purpose, and their signed consent was obtained. 

Each FGD was performed with homogeneous groups to encourage the participants’ free expression. Thus, four FGDs (one with each participant type) were carried out in each municipality (total number of FGDs = 12). A team of three animators did all the FGDs: one facilitator, one who took notes and records, and one observer/moderator. A flip chart was used to illustrate and summarise the main talking points with the participants. The following four aspects were studied successively for each message: (i) what was understood by participants from the drawings; (ii) what was understood from the message written in Malagasy (local language) when read (for illiterate participants); was the meaning of the drawing/written message clear enough?; (iii) was it relevant, i.e., if distributed, would the card help to improve the caregivers and children’s responsiveness?; (iv) could caregivers put the suggestion into practice?

#### 2.3.3. Qualitative Data Analysis

All data collected in the FGs were processed anonymously. 

Discussions were translated and transcribed into French using word processing software. The recordings served to enrich the notes taken during the FGs. Similar ideas and suggestions from different participants were discussed and grouped together. Finally, the key messages and counselling cards were revised to better reflect the local context. 

## 3. Results 

### 3.1. Traditional Feeding Styles and Porridge and Energy Intakes

#### 3.1.1. Socio-Demographic and Anthropometric Characteristics of Participants 

The main characteristics of the 101 child-caregiver pairs enrolled in the study and their families are presented in Table 1. Caregivers (mean age: 26.2 ± SD years) were mostly the mothers (94%), followed by grandmothers and fathers. Their main activity was agriculture, livestock and handicrafts (79% of caregivers). Among the 101 children (9 to 11 months of age), many already had some form of malnutrition: stunted growth, underweight, and wasting were observed in 36.6%, 26.7%, and 7.9% of infants, respectively.

#### 3.1.2. Breastfeeding and Complementary Feeding Practices

According to the mothers’ statement at inclusion, all children were still breastfed at the survey time (Table 2): 57% of mothers said they had exclusively breastfed their child for the first 6 months, and 21.8% said that their child had started to consume porridge or solid food before the age of 6 months.

The children’s diet mainly consisted of cereals (especially rice). Only 23.8% of children had eaten an animal source food on the previous day, and flesh foods and eggs were the least consumed food groups (only 7.9 and 2.0% of children, respectively). The mean diet diversity score, calculated as recently recommended by WHO and UNICEF [36], was 4.0 ± 1.2. More than half of children had a score below the cut-off of five and should be considered at risk for micronutrient deficiency. 

#### 3.1.3. Characteristics of Mealtime, Feeding Practices, and Feeding Behaviours

The porridges prepared by the field workers had a mean DM content of 22.9 ± 1.5 g DM/100 g (corresponding to a high energy density: ~92 kcal/100 g porridge), close to the target (25.0 g DM/100 g porridge) and with low variability (Table 3). The mean duration of the observed meal was 10.8 ± 3.9 min, and less than 10 min for 50% of children. The end of the meal was most often decided by the child (86%) by refusing to eat more. The mean amount of porridge eaten by the children was 98.5 ± 55.8 g (i.e., 12.8 ± 7.5 g/kg BW). Only three children finished all the porridge that was in their bowl, an amount almost equivalent to 30 g/kg BW, the gastric capacity used as reference by WHO [7]. The mean energy intake during the meal (excluding breastmilk) was 90.1 ± 51.4 kcal, i.e., less than a third of the 300 kcal recommended from complementary foods in this age group. Almost half of children had an energy intake below 75 kcal meaning that more than four meals of such energy-dense porridge per day would be required to fully satisfy their energy requirements. According to the mothers, 46% of children had the same appetite as usual, 36% less appetite mainly due to illness (e.g., cough, cold, flu, fever or diarrhoea), and 18% had more appetite than usual. 

The frequency of the different feeding behaviours was determined by coding the videos (Table 4). Positive social and active feeding behaviours were frequently observed (i.e., more than four times during the meal) in 58.4% and 82.2% of caregivers. Positive RF was observed in almost 50% of caregivers, but was rarely frequent. Distraction, whether positive or negative, was not often observed, meaning that most mothers and infants were focused on the meal. Self-feeding was hardly ever used, by mothers and children. Whatever the behaviour type, negative behaviours, leading to a decrease in food intake, were rarely observed (<20% of caregivers). Overall, the “laissez faire” style was largely predominant (68.3% of child-caregiver pairs), showing that mothers primarily let the children decide the meal course. 

#### 3.1.4. Associations with Porridge and Energy Intakes

Potential associations of the main household sociodemographic characteristics, child nutritional status parameters, meal characteristics and feeding behaviours with porridge and energy intake were first investigated by univariate analysis (Table 5). None of the household and infant characteristics investigated was significantly associated with the energy intake from the observed meal (*p* ≥ 0.05). No significant association was found between food intake and mother’s age (*p* = 0.073) and education level (*p* = 0.222), household socioeconomic level (*p* = 0.226), and child’s sex (*p* = 0.961), age (*p* = 0.411) or length (*p* = 0.069). However, when expressed per kg of body weight, porridge intake was negatively associated with the child’s weight, and the weight-for-length, length-for-age and weight-for-age Z-scores (Table 5). Therefore, underweight children ate slightly more than normal-weight children. Concerning mealtime characteristics, porridge intake was strongly and positively associated with the meal duration, and to a lesser extent, with the child’s usual appetite, as assessed by the mothers. Porridge consistency, breastfeeding during the meal or in the hour before, or eating other foods in the hour before had no significant effect on the porridge intake. The child’s health status on the meal day was almost associated with the energy intake (*p*-value = 0.05) that was lower in sick children. After adjustment, three variables (weight-for-age Z-score, meal duration, and usual appetite) remained significantly associated with porridge intake, and two variables (meal duration and usual appetite) with energy intake (Table 6). 

Among the different caregiver’s behaviours, positive RF and social behaviours were positively associated with energy intake (Table 5). Positive RF was also significantly associated with food intake (*p* = 0.029), while social behaviour showed only a trend (*p* = 0.056). The sum of positive behaviours was also significantly associated with food/energy intake, but not the sum of negative behaviours and each individual caregivers’ negative feeding behaviour, consistent with the fact that mothers rarely used them. After adjustment on all significant caregivers’ behaviours, only the positive association between positive RF and porridge and energy intake remained significant (*p* = 0.029 and *p* = 0.027 respectively) (Table 7). 

In children, positive behaviours were significantly associated with increased porridge and energy intake, resulting in a significant positive effect of the sum of their positive behaviours. Among their negative behaviours, only negative RF showed a significant association with decreased intake. Moreover, the sum of negative behaviours was negatively associated with food and energy intake. Self-feeding behaviours (negative or positive, and by the mother or child) were rare and were not associated with porridge/energy intake. Finally, the overall feeding style also was associated with porridge and energy intakes. Three associations between infants’ feeding behaviour and intakes remained significant after adjustment: the positive association of porridge and energy intake with positive active feeding and distraction (*p* ≤ 0.001), and the negative association of the sum of negative behaviours with the two intakes (*p* < 0.001) (Table 7).

### 3.2. Development of Key Messages and an Awareness-Raising Tool to Promote Responsive Feeding and Validation in the Community

From the frequency of use of positive feeding behaviours obtained from the analysis of mealtime videos (Table 4) and by comparison with the recommendations by PAHO [24] and Moore et al. [31], six key messages were formulated to promote RF practices that could lead to positive mealtime outcomes. Each message was accompanied by two to three recommended practices corresponding to specific situations (Table 8). When looking at the frequency of use of the different feeding behaviour types, positive self-feeding and distraction were not used/rarely used by 90% and 75% of caregivers, respectively. Therefore, the key messages 4 and 6 were formulated to promote the use of these two positive behaviours that can help to increase the children’s interest in eating. To reinforce the use of positive RF behaviours, which were used very frequently only by 7% of caregivers, the key messages 1, 2 and 3 were formulated to make mealtime a time of pleasant exchange between caregivers and children, focused on the consumption of complementary foods. The key messages 5 and 6 stress the importance of verbal exchanges, including encouragement and praise. Counselling cards illustrating these messages (Figure 1) adapted to the local context (e.g., colour and type of clothing, home environment, landscape) were developed as a promotion tool. 

### 3.3. Assessment of the Key Messages’ Acceptability by the Local Community 

In total, 98 participants (mothers, fathers, other caregivers, and CHW) were involved in the 12 FGs organised in three different areas of the district (Table 9). Most participants were farmers. There were differences in education level among areas. For instance, the number of mothers with low education level was higher in the rural area further away from the district capital (i.e., municipality 3). 

None of the six messages was rejected. All participants found the six messages important and relevant. Although these are common practices and common sense, participants were aware that it was important to repeat them for improving the caregivers’ knowledge and reducing inappropriate practices and behaviours, especially among first-time mothers. Improvements were suggested about the formulation of the messages/practices and the drawings to facilitate their understanding by the community and avoid confusion. The suggestions, approved by most participants, led to slight modifications of the formulation of the written messages and/or the corresponding drawing (Figure 1). 

Message 1: Most participants noticed the hug between caregiver and child, but only few said that they were looking at each other and smiling. They suggested swapping the two drawings, because they thought more logical that the mother should show her love to the child to put him/her in a good mood to eat. According to a father of municipality 2, the mother’s hug means “*When you eat well, I love you, child*”. According to other caregivers from municipality 3, the image suggests saying words such as “eat your meal” or “it is good”.

Message 2: Almost all participants understood the message that there is a time for breastfeeding and a time for feeding, but the interval between breastfeeding and the next solid meal generated a lot of discussion. Finally, all agreed that 30 min would be an acceptable and feasible interval and they suggested increasing the clock size to improve its visibility. 

Message 3: From the drawing alone, most participants misunderstood the message, thinking that the child is happy and rushes to the meal, although the recommendation was to leave the child’s arms free when feeding, to avoid feeling constrained. This led to a modification of the drawing.

Message 4: All participants clearly understood from the image that it deals with a child learning to use the spoon and self-feed. The main message was slightly reworded based on the participants’ suggestions.

Message 5: The clock in the images put some participants on the wrong track and was removed. Most participants clearly understood that when the bowl is empty, this means that the child has eaten all the food offered and the caregiver congratulates the child by clapping the hands and saying “*Bravo, my child eats very well*”. A CHW from village 2 also said “*The caregiver feels happy because she has reached her goal*”. On the other hand, the message about encouraging the child to eat was not recognised until after the written message was read.

Message 6: All participants understood that the idea was to distract the child during the meal by giving a toy to handle or by telling stories or singing; however, this raised some discussion. Some participants said “*The children may not eat if toys are given at mealtime. It is best to give toys after the meals*”. The musical notes were not recognised by most participants from the remote rural village 3 and were removed from the image.

Finally, most participants found the messages and practices applicable; however, they stressed that the main obstacles to their implementation were the little time available to mothers and the need to leave the child to a caregiver to go to work.

## 4. Discussion

The mean amount of food ingested by children during one meal taken at home in the usual conditions was low and highly variable (12.8 ± 7.5 g porridge/kg BW/meal), indicating that many children had a small appetite, well below the assumed gastric capacity of 30 g food/kg BW/meal [7]. This situation is commonly observed in similar contexts where chronic malnutrition is highly prevalent [8,9,10], whatever the quality of the food offered. In these conditions, only few children might satisfy their nutrient and energy requirements from complementary foods, estimated at 300 kcal/day in the 9–11-month age group [20]. Moreover, the diet diversity score was below five (out of eight food groups) for 89.1% of children, indicating an increased risk of micronutrient deficiencies that may impair the child’s development [36]. In these conditions, at each meal, children are at high risk of repeated deficits in energy and nutrient intake. This can be related to the children’s nutritional status in the study sample with a very high prevalence of stunting for this age group (36.6%), consistent with national survey data in this region [4]. The significant negative association between weight-for-age Z-score and porridge intake, expressed per kg of BW, in our study suggests that underweight children might eat larger amounts of food per kg of BW than normal-weight children. Such association was reported by several previous studies in young children [37,38,39]. Meal duration also was strongly associated with food and energy intake, highlighting the need to allow the child sufficient time to eat. This was also shown in a study in Burkina Faso where mothers were invited to offer food to their child a second and a third time after they had declared the meal over [8]. 

The analysis of the feeding behaviours, identified according to the classification by Moore et al. [31], showed that negative behaviours (i.e., decreasing the food intakes) were rarely used by caregivers in the present study. In a previous study in Ethiopia, behaviours with negative effect on food intake, such as force-feeding, were more frequent [29]. However, the use of positive behaviours should be strengthened because they were observed only in about half of caregivers, and at most two to four times per meal. In the few cases where caregivers used positive RF behaviours more often during the study meal (*n* = 7), the mean food intake was strongly increased (+75%) (Table 7). This suggests that promoting this behaviour type could improve the nutritional intake by children, especially those with small appetite. Such effect was already reported by Aboud et al., (2009) [27], but was not precisely quantified (estimated through mouthful count). Another study recently proposed a more comprehensive RF behaviour rating scale, but further investigation is needed to validate and quantify the link between these ratings and children dietary intake [30].

The counselling cards were based on six simple messages focused on the caregiver–child interactions during a meal taken separately from the rest of the family. All messages were validated by caregivers and also by CHW involved in community development projects. Similarly, in a recent study in Ghana, other RF messages focused on different aspects of feeding were all validated by the community [40]. Indeed, these messages are mainly based on common sense. In Madagascar, several of these messages are already promoted by the national Infant and Young Child Feeding policy. However, they are often buried in a multitude of recommendations and sometimes do not hold the attention of community workers who do not see the value of their specific promotion. Therefore, although people often find it difficult to understand the messages from the illustrations alone, all participants agreed that it is essential to accompany such messages with culturally appropriate images and to use the counselling cards as a memo, including for people who cannot read, especially in rural areas. 

The FG participants also highlighted that implementing these messages requires patience by the person feeding the child. For mothers, who are often responsible for multiple household tasks, time can really be the main limiting factor. The involvement of other caregivers than the mother could provide a partial solution. However, awareness raising about each message remains essential. The question is to determine which actor could take charge of this awareness-raising activity within the community.

## 5. Conclusions

This study highlights the very low food intakes of many young children at mealtimes in rural villages of the Amparafaravola district in Madagascar, as often observed in settings where stunting is highly prevalent. However, it also identifies local RF practices that promote a significant increase in food/energy intake. A long-term increase of food/energy intake could allow young children to regain a normal appetite for their age, thus improving the satisfaction of their nutritional requirements and subsequently their nutritional status. Including awareness of RF practices in strategies to promote nutritionally adequate food and good hygiene and care practices could create conditions more conducive to catch-up growth. The value of such integrated strategies needs to be investigated in longitudinal studies in similar settings.

## Figures and Tables

**Figure 1 nutrients-14-00361-f001:**
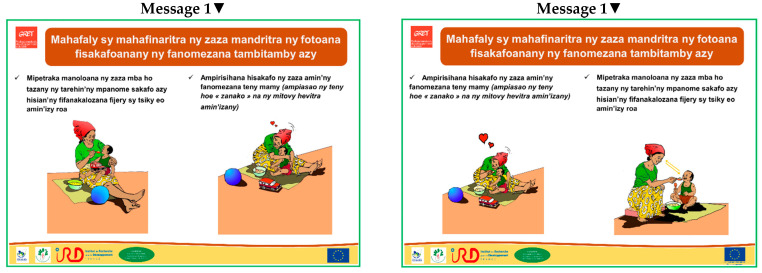
Counselling cards illustrating the six key messages to promote responsive feeding before (**left panels**) and after (**right panels**) validation by the community.

**Table 1 nutrients-14-00361-t001:** Main characteristics of households, caregivers and children (*n* = 101).

	Frequency (%)	Mean ± SD
**Socio-demographic characteristics**		
Mother’s age (year)		26.2 ± 7.3
Mother’s occupation		
Employees	6.9	
Self-employed (agriculture, livestock, crafts, …)	79.2	
Without income	13.9	
Mother’s education		
No school	8.9	
Primary school	60.4	
Secondary school and more	30.7	
Number of people per living room		2.9 ± 1.5
Infant’s age (months)		10.3 ± 0.9
Sex		
Boys	51.5	
Girls	48.5	
Birth order		2.4 ± 1.5
**Infants’ nutritional status**		
Weight at birth (kg), *n* = 59		3.15 ± 0.61
Low birth weight (≤2.5 kg)	15.3	
Weight (kg)		7.71 ± 1.02
Length (cm)		68.5 ± 2.7
Weight-for-Length Z-score ^a^		−0.45 ± 1.09
Length-for-Age Z-score ^a^		−1.61 ± 0.99
Weight-for-Age Z-score ^a^		−1.23 ± 1.08
Wasting	7.9	
Stunting	36.6	
Underweight	26.7	

^a^ WHO growth standard (2006).

**Table 2 nutrients-14-00361-t002:** Breastfeeding and complementary feeding practices of infants (*n* = 101).

	Frequency (%)	Mean ± SD
Exclusively breastfed for the first six months	57.4	
Age of introduction of porridge or solid food (months)		6.2 ± 1.2
<6 months	21.8	
6 months	42.6	
>6 months	35.6	
Breastfed the day before the survey	100	
Infants who ate foods from the following food groups in the previous day		
Cereals, roots and tubers	100	
Vitamin A-rich fruits and vegetables	70.3	
Other fruits and vegetables	53.5	
Legumes and nuts	36.6	
Dairy products	17.8	
Flesh foods	7.9	
Eggs	2.0	
Dietary diversity		
Dietary diversity score ^a^		4.0 ± 1.2
Minimum dietary diversity ^b^	10.9	

^a^ Eight food groups including breastmilk as one group; ^b^ Minimum dietary diversity: percentage of infants who received at least five among the eight food groups.

**Table 3 nutrients-14-00361-t003:** Feeding practices before/during the study meal (*n* = 101).

Variables	Frequency (%)	Mean ± SD
Children breastfed after waking up in the morning	72.3	
Children ate something else before the study meal	40.6	
Children breastfed during the study meal	39.6	
Porridge DM content (g DM/100 g)		22.9 ± 1.5
Porridge energy density (kcal/100 g)		91.6 ± 6.0
Porridge intake (g/meal)		98.5 ± 55.8
Porridge intake per kg body weight (g/kg BW/meal)		12.8 ± 7.5
<10 g/kg BW/meal	47.5	
10–20 g/kg BW/meal	33.7	
20–30 g/kg BW/meal	15.8	
≥30 g/kg BW/meal	3.0	
Energy intake (kcal/meal)		90.0 ± 51.4
Energy intake <75 kcal/meal	49.5	
Duration of the meal (min)		10.8 ± 3.9
Number of bites refused		3.7 ± 3.5
Reason for ending the meal		
There was some porridge left, but the child refused to continue eating	86.1	
There was some porridge left, but the caregiver decided that the meal was over	5.9	
There was no more porridge in the child’s bowl	7.9	
Health status of the child on the day of observation		
In good shape	60.4	
Sick	39.6	
Assessment of the child’s usual appetite (mother’s opinion)		
He/she has a small appetite	15.8	
He/she eats normally	24.8	
He/she has a big appetite	59.4	
Appetite of the child during the study meal compared to usual		
Less than usual	35.6	
As usual	46.5	
More than usual	17.8	

DM: dry matter; BW: body weight.

**Table 4 nutrients-14-00361-t004:** Feeding behaviours of caregivers and children during the study meal (*n* = 101).

	% Caregivers		% Children	
Feeding Behaviour	Positive	Negative	Positive	Negative
**Self-feeding**				
Absent (0–1 time)	90.1	85.1	76.2	98.0
Present (2–4 times)	6.9	12.9	17.8	2.0
Frequent (≥5 times)	3.0	2.0	5.9	0.0
**Responsive feeding**				
Absent (0–1 time)	49.5	99.0	13.9	26.7
Present (2–4 times)	43.6	1.0	55.4	56.4
Frequent (≥5 times)	6.9	0.0	30.7	16.8
**Active feeding**				
Absent (0–1 time)	4.0	80.2	5.0	72.3
Present (2–4 times)	13.9	16.8	22.8	27.7
Frequent (≥5 times)	82.2	3.0	72.3	0.0
**Social behaviour**				
Absent (0–1 time)	7.9	83.2	17.8	67.3
Present (2–4 times)	33.7	16.8	53.5	32.7
Frequent (≥5 times)	58.4	0.0	28.7	0.0
**Distraction**				
Absent (0–1 time)	75.2	85.1	63.4	88.1
Present (2–4 times)	20.8	14.9	30.7	11.9
Frequent (≥5 times)	4.0	0.0	5.9	0.0
**Overall feeding style**	Frequency in caregiver–child pairs (%)			
Responsive feeding	6.9			
Active positive	19.8			
Laissez-faire	68.3			
Active negative	5.0			

**Table 5 nutrients-14-00361-t005:** Variables significantly associated with the children’s porridge and energy intake: univariate analysis.

Variables	Porridge Intake g/kg BW		Energy Intake kcal/meal	
	*p*-Value	Coef.	*p*-Value	Coef.
	**General characteristics of households and infants**			
Infant’s weight	**0.020** ^b^	−1.73	0.913 ^b^	−0.56
WLZ	**0.002** ^b^	−2.15	0.069 ^b^	−8.55
LAZ	**0.019** ^b^	−1.77	0.470 ^b^	−3.76
WAZ	**<0.001** ^b^	−2.50	0.080 ^b^	−8.24
	**Characteristics of the mealtime**			
Duration of the feeding episode	**<0.001** ^b^	+0.019	**<0.001** ^b^	+0.122
Usual appetite of the child	**0.037** ^a^	+1.73	**0.020** ^a^	+13.1
	**Caregivers’ feeding behaviours**			
Responsive feeding positive	**0.029** ^b^	+2.62	**0.027** ^b^	+18.1
Social behaviour positive	0.056 ^b^	+2.23	**0.048** ^b^	+ 15.7
	**Sum of caregivers’ feeding behaviours**			
Sum of positive behaviours	**0.010** ^b^	+1.29	**0.016** ^b^	+8.26
	**Infants’ feeding behaviours**			
Responsive feeding positive	**0.033** ^b^	+2.45	**0.031** ^b^	+16.9
Responsive feeding negative	**0.006** ^b^	−3.09	**0.006** ^b^	−21.2
Active feeding positive	**<0.001** ^b^	+5.73	**<0.001** ^b^	+39.1
Social behaviour positive	**0.011** ^b^	+2.80	**0.006** ^b^	+20.6
Distraction positive	**0.008** ^b^	+3.26	**0.012** ^b^	+21.0
	**Sum of infants’ behaviours**			
Sum of positive behaviours	**<0.001** ^b^	+2.22	**<0.001** ^b^	+14.8
Sum of negative behaviours	**0.017** ^b^	−1.45	**0.012** ^b^	−10.3
Overall feeding style	**0.031** ^a^		**0.023** ^a^	
Laissez-faire				
Active negative		−3.84		−25.8
Active positive		+0.55		5.15
Responsive feeding		+5.55		37.3

Bold type indicates significant associations (*p* < 0.05); (a) one-way analysis of variance (ANOVA); (b) simple linear regression. No significant association with food or energy intake (*p* > 0.05) was found for the following variable: mother’s education level or age; household socio-economic score; infant’s sex, age, and length; porridge consistency; child’s health status on the observation day; breastfed during the meal; breastfeeding or complementary food feeding in the hour before the porridge meal; positive or negative self-feeding behaviour by the caregiver or the child; all negative feeding behaviours by the caregiver; positive active feeding, social behaviour and distraction behaviours by the caregiver; negative active feeding, social behaviour and distraction by the caregiver. WLZ: weight-for-length Z-score; LAZ: length-for-age Z-score; WAZ: weight-for-age Z-score.

**Table 6 nutrients-14-00361-t006:** Identification of variables with a significant effect on the children’s intakes: multivariate analysis ^1^.

Variable	Porridge Intake g/kg BW *p*-Value Mean ± SD	Energy Intake kcal/meal *p*-Value Mean ± SD
Caregiver’s age (years)	NS	X ^2^
Child length (cm)	NS	X ^2^
Weight-for-length Z-score	NS	NS
Length-for-age Z-score	NS	X ^2^
Weight-for-age Z-score (WAZ)	*p* < 0.001	NS
Underweight (WAZ < −2ET) (*n* = 27)	18.0 ± 8.6	
Not underweight (WAZ ≥ −2ET) (*n* = 74)	10. 9 ± 6.1	
Meal duration	*p* < 0.001	*p* < 0.001
≤8.6 min (*n* = 36)	8.3 ± 4.1 ^a^	61.1 ± 30.2 ^a^
8.7–11.9 min (*n* = 33)	11.6 ± 6.1 ^b^	82.0 ± 46.0 ^b^
≥12.0 min (*n* = 32)	19.0 ± 7.7 ^c^	130.5 ± 49.8 ^c^
Child’s health status on the observation day	NS	X ^2^
Usual child’s appetite as declared by mothers	*p* = 0.024	*p* = 0.024
Small appetite (*n* = 16)	9.5 ± 8.5 ^a^	64.4 ± 43.3 ^a^
Normal appetite (*n* = 25)	11.0 ± 4.8 ^ab^	78.7 ± 30.3 ^ab^
Big appetite (*n* = 60)	14.4 ± 7.8 ^b^	101.4 ± 56.6 ^b^

SD: standard deviation; NS: not significant (*p*-value ≥ 0.05); ^1^ using the generalised linear model procedure; X ^2^: the variable was not entered in the model because not significant in the univariate analysis (see Table 6). In each column different superscript letters indicate significant difference between adjusted means at *p* < 0.05.

**Table 7 nutrients-14-00361-t007:** Feeding behaviours significantly associated with the children’s porridge and energy intakes and adjusted means: multivariate analysis.

Feeding Behaviour Type	Porridge Intake g/kg BW *p*-Value Mean ± SD	Energy Intake kcal/meal *p*-Value Mean ± SD
**Caregivers’ feeding behaviour**		
Self-feeding, positive	NS	x
Responsive feeding, positive	*p* = 0.029	*p* = 0.027
Absent (*n* = 50)	11.9 ± 7.0 ^a^	82.9 ± 48.6 ^a^
Present (*n* = 44)	12.58 ± 7.0 ^ab^	90.0 ± 50.6 ^a^
Frequent (*n* = 7)	20.70 ± 10.1 ^c^	139.8 ± 50.6 ^b^
Social behaviour, positive	x	NS
Sum of the caregivers’ positive behaviours	NS	NS
**Infants’ feeding behaviour**		
Self-feeding, positive	NS	x
Responsive feeding, positive	NS	NS
Responsive feeding, negative	NS	NS
Active feeding, positive	*p* < 0.001	*p* < 0.001
Absent (*n* = 5)	3.1 ± 0.99 ^a^	24.0 ± 7.5 ^a^
Present (*n* = 23)	9.0 ± 5.2 ^b^	63.8 ± 39.5 ^b^
Frequent (*n* = 73)	14.7 ± 7.4 ^c^	102.7 ± 49.6 ^c^
Social behaviour, positive	NS	
Distraction, positive	*p* < 0.001	*p* = 0.001
Absent (*n* = 64)	11.7 ± 6.3 ^a^	82.4 ± 44.6 ^a^
Present (*n* = 31)	13.6 ± 8.5 ^ab^	96.2 ± 60.2 ^ab^
Frequent (*n* = 6)	20.8 ± 9.8 ^b^	137.8 ± 41.4 ^b^
Sum of the infants’ positive behaviours	NS	NS
Sum of the infants’ negative behaviours	*p* < 0.001	*p* < 0.001
Absent (*n* = 21)	16.8 ± 9.9 ^b^	118.6 ± 67.7 ^b^
Present (*n* = 27)	12.9 ± 6.2 ^ab^	91.1 ± 43.4 ^ab^
Frequent (*n* = 53)	11.2 ± 6.6 ^a^	78.0 ± 42.8 ^a^

x: the variable was not entered in the model because not significant in univariate analysis (see Table 6). NS: not significant (*p*-value ≥ 0.05). In each column different superscript letters indicate significant differences between adjusted means at *p* < 0.05.

**Table 8 nutrients-14-00361-t008:** Six key messages proposed for the promotion of responsive feeding and the corresponding examples of recommended practices.

Key Message	Examples of Recommended Practices
Key message 1 To allow the child to enjoy the time spent eating, make the mealtime a time for tender exchanges	Example 1: Turn your face towards the child during meals to allow the child to see your face, to promote eye-to-eye contact and smile exchanges.
	Example 2: Encourage the child to eat with tender words (use the word “my baby” or equivalent).
Key message 2 To increase the child’s appetite during meals, separate the moments of breastfeeding and feeding	Example 1: Wait for a while after breastfeeding before offering complementary food to your child.
	Example 2: Breastfeed the child outside the mealtime.
Key message 3 To increase the child’s appetite, avoid force-feeding	Example 1: Leave the child’s arms free when feeding.
	Example 2: Only put food in the child’s mouth when the child agrees to eat, avoid putting food in the mouth of a crying child.
Key message 4 To increase the child’s appetite, encourage self-feeding	Example 1: Give the child a second spoon during the meal so that the child can try to use it.
	Example 2: Help the child to hold the spoon and guide the child’s hand to the mouth when the child tries eating alone.
Key message 5 To increase the child’s interest in food, praise your child when eating well and encourage your child when not eating well	Example 1: Use words or signs of congratulation well-known by the child (bravo, hand clapping, etc.).
Key message 6 To allow the child to enjoy the meal, grab your child’s attention and distract your child	Example 1: Talk to your child.
	Example 2: Sing a song known by the child.
	Example 3: Give to your child a clean colourful toy to handle.

**Table 9 nutrients-14-00361-t009:** Main characteristics of the focus group participants.

	All	Municipality 1 (with Health Centre 2)				Municipality 2 (with Health Centre 2)				Municipality 3 (with Health Centre 1)			
		FG1	FG2	FG3	FG4	FG5	FG6	FG7	FG8	FG9	FG10	FG11	FG12
		Mother	Father	CHW ^a^	OC ^b^	Mother	Father	CHW	OC	Mother	Father	CHW	OC
	(*N* = 98)	(*n* = 5)	(*n* = 4)	(*n* = 9)	(*n* = 3)	(*n* = 10)	(*n* = 7)	(*n* = 8)	(*n* = 10)	(*n* = 15)	(*n* = 11)	(*n* = 7)	(*n* = 9)
Mean age (years)	38.6 ± 13.1	29.8 ± 5.8	32.5 ± 9.0	47.7 ± 2.7	53.7 ± 10.1	24.1 ± 3.9	35.4 ± 3.4	51.5 ± 10.3	49.6 ± 12.2	26.2 ± 7.5	29.8 ± 4.9	45.6 ± 4.5	52.7 ± 8.6
**Sex** (%)													
Women	72	100	-	0	100	100	-	37	100	100	-	29	100
Men	28	-	100	100	-	-	100	63	-	-	100	71	-
**Main activity** (%)													
Self-employed	95	100	75	89	100	100	100	88	90	93	100	100	100
Temporary job	2	0	25	0	0	0	0	0	10	0	0	0	0
No regular income	3	0	0	11	0	0	0	12	0	7	0	0	0
Permanent job	0	0	0	0	0	0	0	0	0	0	0	0	0
**Education level** (%)													
No school	9	0	50	0	33	0	0	0	0	13	9	0	33
Primary school	38	0	50	67	67	20	0	0	10	80	72	0	56
Secondary school	50	100	0	22	0	80	72	100	90	7	18	100	11
Higher education	3	0	0	11	0	0	28	0	0	0	0	0	0

^a^ Community health workers; **^b^** other caregivers.

## Data Availability

Data cannot be shared because we are committed to data confidentiality (ethical issues).

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
