# Peer review of "Observation of Traditional Caregiver-Infant Feeding Behaviours and Porridge and Energy Intakes during One Meal to Define Key Messages for Promoting Responsive Feeding in the Amparafaravola District, Rural Madagascar"

_nutrients, 2022, doi:10.3390/nu14020361_

Round 1

Reviewer 1 Report

This article reports on a study involving caregiver and infant dyads and the assessment of malnutrition among infants in Madagascar as well as ways to promote an increase in food/energy intake among a child population known for experiencing chronically malnutrition. While the topic is quite important, the manuscript is not yet publishable in its current form.

Introduction

the introduction is describes well the current situation of malnutrition and is engaging.

Methods

In the methods section, the authors discuss the inclusion criteria of the infant child but there is no mention of the caregiver eligibility criteria. There also seem to be some typos in this sentence “the main inclusion criteria were: BE 9-11-month-old, be breastfed, have [109??] started to consume complementary foods at the time of the survey, and not suffer from severe wasting 110 (Z-score weight-for-length > -3).

There is little information on the questions or measures used in the cross-section survey (it seems the survey is really a socio-demographic questionnaire) this needs to be clarified and if a full survey more explanation is needed regarding the questions asked, if scales were used, etc.

Was the socio demographic data collected with the caregiver and did it include information about the caregiver and infant? This should be clarified.

For the focus groups, were participants from different backgrounds (e.g,. CHWs, parents) are put together in focus groups or were the focus groups separates by stakeholder group. This should be clarified and a justification provided.

The analysis section of the qualitative research is not well described. Was it an inductive or deductive approach to analysis? how were themes or messages identified?

Results

The analysis of the focus groups are not presented in a meaningful way. What were the six messages? What were the themes or patterns that emerged that informed these messages.  

Reviewer 2 Report

A novel well designed multistage study.

Suggestion for change- Re-consider  the generalization statement (386) in view of the study design, study group ( infants 9 -11 months) and study location (one district of Malaysia).

Author Response

Many thanks for your positive appreciation of our work. Our study was indeed carried out in the Amparafaravola district of Madagascar.

Following your reommendation, the conclusion was changed as follows:

This study highlights the very low food intakes of many young children at mealtimes in rural villages of the Amparafaravola district in Madagascar, as it is often observed in settings where stunting is highly prevalent. However, it also identifies local RF practices that promote a significant increase in food/energy intake. A long-term increase of food/energy intake could allow young children to regain a normal appetite for their age, thus improving the satisfaction of their nutritional requirements and subsequently their nutritional status. Including awareness of RF practices in strategies to promote nutritionally adequate food and good hygiene and care practices could create conditions more conducive to catch-up growth. The value of such integrated strategies needs to be investigated in longitudinal studies in similar settings.